# Hepatic AMPK signaling dynamic activation in response to REDOX balance are sentinel biomarkers of exercise and antioxidant intervention to improve blood glucose control

**Meiling Wu[1†], Anda Zhao[1†], Xingchen Yan[1†], Hongyang Gao[2†], Chunwang Zhang[1], Xiaomin Liu[1], Qiwen Luo[1], Feizhou Xie[3], Shanlin Liu[4,5]\*, Dongyun Shi[1,4]\***

[1]Key Laboratory of Metabolism and Molecular Medicine of the Ministry of Education, Department of Biochemistry and Molecular Biology, School of Basic Medical Sciences, Fudan University, Shanghai, China; [2]Institute of Electronmicroscopy, School of Basic Medical Sciences, Fudan University, Shanghai, China; [3]Changning Maternity and Infant Health Hospital, East China Normal University, Shanghai, China; [4]Free Radical Regulation and Application Research Center of Fudan University, Shanghai, China; [5]Department of Biochemistry and Molecular Biology, School of Basic Medical Sciences, Fudan University, Shanghai, China

**\*For correspondence:**
slliu@shmu.edu.cn (SL);
dyshi@fudan.edu.cn (DS)

[†]These authors contributed equally to this work

**Competing interest:** The authors declare that no competing interests exist.

**Abstract** Antioxidant intervention is considered to inhibit reactive oxygen species (ROS) and alleviate hyperglycemia. Paradoxically, moderate exercise can produce ROS to improve diabetes. The exact redox mechanism of these two different approaches remains largely unclear. Here, by comparing exercise and antioxidant intervention on type 2 diabetic rats, we found moderate exercise upregulated compensatory antioxidant capability and reached a higher level of redox balance in the liver. In contrast, antioxidant intervention achieved a low-level redox balance by inhibiting oxidative stress. Both of these two interventions could promote glucose catabolism and inhibit gluconeogenesis through activation of hepatic AMP-activated protein kinase (AMPK) signaling; therefore, ameliorating diabetes. During exercise, different levels of ROS generated by exercise have differential regulations on the activity and expression of hepatic AMPK. Moderate exercise-derived ROS promoted hepatic AMPK glutathionylation activation. However, excessive exercise increased oxidative damage and inhibited the activity and expression of AMPK. Overall, our results illustrate that both exercise and antioxidant intervention improve blood glucose control in diabetes by promoting redox balance, despite different levels of redox state(s). These results indicate that the AMPK signaling activation, combined with oxidative damage markers, could act as sentinel biomarkers, reflecting the threshold of redox balance that is linked to effective glucose control in diabetes. These findings provide theoretical evidence for the precise management of diabetes by antioxidants and exercise.

## Editor's evaluation

Redox signaling is a dynamic and concerted orchestra of interconnected cellular pathways. There is always a debate whether ROS (reactive oxygen species) could be a friend or foe. There are several paradoxical studies (both animal and human) wherein exercise health benefits were reported to be accompanied by increases in ROS generation. Utilizing the in-vitro studies as well as rats models, this

manuscript illustrates the different regulatory mechanisms of exercise and antioxidant intervention on redox balance/redox state(s) that are linked to improved glucose control and thereby effective management of diabetes.

## Introduction

Diabetes mellitus is a chronic metabolic disease, that has emerged as a global public health problem. According to the latest epidemiological data from the International Diabetes Federation, the global diabetes prevalence in 20–79 year-old was estimated to be 10.5% (536.6 million people) in 2021, and is expected to rise to 12.2% (783.2 million) in 2045 (**Sun et al., 2022**). With the development of genomics, proteomics, and metabolomics, it has been discovered by many studies that type 2 diabetes is associated with irreversible risk factors such as age, genetics, race, and ethnicity and reversible factors such as diet, physical activity, and lifestyle (**Heald et al., 2020**; **Chan et al., 2021**).

Aerobic metabolism in glucose oxidation, mitochondrial damage, and oxidative stress have been considered to play a critical role in the occurrence and development of diabetes (**Iacobini et al., 2021**). Exercise and antioxidant supplements are often suggested as essential therapeutic strategies in the early stages of type 2 diabetes (**Kirwan et al., 2017**; **Bhatti et al., 2022**), with different mechanisms. It has been reported that chronic exercise training can alleviate oxidative stress and diabetic symptoms by improving cellular mitochondrial function and biogenesis in the diabetic state (**Stanford and Goodyear, 2014**). Contradictorily, exercise also increases reactive oxygen species (ROS) production, while prolonged or high-intensity exercise could result in mitochondrial functional impairment to aggravate complications of diabetes (**Flockhart et al., 2021**). Since 1970s, studies have demonstrated that 1 hr of moderate endurance exercise can increase lipid peroxidation in humans (**Brady et al., 1979**; **Dillard et al., 1978**). In 1998, Ashton directly detected increasing free radical levels in exercising humans using electron paramagnetic resonance spectroscopy (EPR) and spin capture (**Ashton et al., 1998**). These results led to a great deal of interest in the role of ROS in physical exercise (**Powers and Jackson, 2008**; **Powers et al., 2011**; **Traverse et al., 2006**). Regarding the contradiction of exercise on ROS scavenging or production, James D Watson also hypothesized that type 2 diabetes is accelerated by insufficient oxidative stress rather than oxidative stress (**Watson, 2014**), based on the effect of exercise on diabetes management. Although Watson's opinions supported that exercise could treat diabetes by producing ROS, whether exercise-induced ROS production is beneficial or detrimental to diabetes is still being debated. The specific regulation of ROS produced by exercise on diabetic blood glucose *in vivo* is unclear. In contrast, the general view of the antioxidant treatment for diabetes is that antioxidants reduce cytotoxic ROS and oxidative products, thus alleviating diabetes and achieving glycemic control (**Rahimi et al., 2005**). Our previous study also found that hepatic mitochondrial ROS scavengers and antioxidant substances inhibited the oxidative products such as Malondialdehyde (MDA) and 4-HNE in diabetic animals and favored glycemic control (**Wu et al., 2019**; **Wu et al., 2021**). Exercise-induced oxidation and antioxidant administration, as two opposite approaches, could achieve the regulation of diabetes, respectively. However, the differences in redox mechanisms between these two approaches to diabetes treatment have not been fully understood.

It is well established that the increase of skeletal muscle glucose uptake during exercise is crucial in glycemic control (**Holloszy, 2005**; **Holloszy et al., 1986**; **Greiwe et al., 1999**). Considering that the liver is another vital organ for maintaining blood glucose homeostasis, including storing, utilizing, and producing glucose, exercise-induced hepatic redox metabolism is also significant. The activation of hepatic AMP-activated protein kinase (AMPK), which acts as a 'metabolic master switch', alleviates diabetes symptoms by reducing glycogen synthesis, increasing glycolysis, and promoting glucose absorption in surrounding tissues (**Viollet et al., 2006**). Therefore, the activation of AMPK in the liver is significant for regulating glucose and lipid metabolism in the blood. Zmijewski et al. found that AMPK could be activated by hydrogen peroxide stimulation through direct oxidative modification (**Zmijewski et al., 2010**). In contrast, other studies suggested that oxidative stress could disrupt the activation of the AMPK signaling pathway (**Ren et al., 2021**; **Hawley et al., 2010**). Our previous study explored the mechanism by which redox status contributes to hepatic AMPK dynamic activation. Under a low ROS microenvironment, GRXs mediated S-glutathione modification activates AMPK to improve glucose utilization. In contrast, under an excessive ROS microenvironment, sustained high level ROS might cause loss of AMPK

**eLife digest** Molecules known as reactive oxygen species or ROS play vital roles in healthy cells. However, ROS can act as a double-edged sword: if their levels become too high, they can be harmful and interfere with many physiological processes. Indeed, diabetes, high blood pressure and many other chronic diseases are associated with imbalances in the levels of ROS in the body. To counter high ROS levels, cells have antioxidant mechanisms that reduce the excess ROS in the cell and keep the 'redox' (from reduction and oxidation) balance of the cell.

Exercise and antioxidant nutritional supplements have attracted much attention as drug-free interventions for diabetes. Both strategies alter the levels of ROS in the body, with exercise increasing the levels of ROS, and antioxidant supplements reducing them. Individuals with diabetes and other metabolic health issues have different ROS levels depending on the severity of the disease, age, genetics and other factors, leading to different redox states in their cells. Thus, approaches that can accurately evaluate the redox balance status of individuals are necessary for clinicians to identify what types of exercise and antioxidant supplements are beneficial and which treatments are most appropriate for each patient.

Wu, Zhao, Yan, Gao et al. examined the effects of exercise and antioxidant supplements on rats with diabetes, with the aim of identifying molecules – also known as biomarkers – that reflect the bodies' redox balance. They found that moderate exercise increased the levels of ROS in the liver, which, in turn, compensated by increasing the production of antioxidants to protect against the higher levels of ROS. This resulted in a healthy 'high-level' redox balance, in which both ROS and antioxidants levels were high in the rats. On the other hand, giving the rats antioxidant supplements decreased their levels of ROS, leading to a healthy low-level redox balance with low levels of ROS.

These findings indicate that regular moderate exercise may be appropriate for people with pre-diabetes symptoms to restore a healthy redox balance. This is because the compensatory antioxidant mechanisms that kick in during exercise may be enough to counteract the excessive levels of ROS in these people. For patients with mild diabetes, exercise, antioxidant supplements, or a combination of both may be appropriate treatment, depending on their levels of ROS. Finally, patients with severe diabetes, who already have high levels of ROS, may benefit from antioxidant supplements to help reduce their excessive levels of ROS.

In the future, the biomarkers identified by Wu, Zhao, Yan, Gao et al. may be used to monitor and assess the change in the redox balance status of various populations and guide personalized interventions to maintain health. Additionally, these findings provide a new strategy for precision prevention and treatment of diabetes and other metabolic diseases.

---

protein (*Dong et al., 2016b*). These studies indicate that oxidative modification can directly regulate AMPK activity in liver cells, thus activating downstream signaling pathways to regulate glucose and lipid metabolism. However, it is unclear why two seemingly contradictory phenomenon of antioxidant intervention and exercise-induce ROS can promote AMPK activation. Moderate exercise has been proved significantly elevate systemic ROS. At the same time, endogenous antioxidant defences also increased to counteract increased levels of ROS induced by exercise (*Parker et al., 2014*). Thus, we hypothesized that both antioxidants and exercise could reach either high- or low-level redox balance in diabetic individuals. Moreover, the activity and expression of AMPK might be a biomarker of redox balance *in vivo*.

Hence, the present study was designed to understand the different mechanisms of exercise and antioxidant intervention in diabetes and verify the activation of hepatic AMPK as a hallmark of dynamic redox balance. First, we utilized the streptozotocin-high fat diet (STZ-HFD) induced type 2 diabetic model (T2DM) in rats to clarify the hepatic redox status in T2DM rats after the exercise or antioxidant intervention. Then, according to the exercise intensity and mode, we divided the exercise groups into three modes and found that AMPK activation could serve as a sentinel biomarker of redox balance and moderate exercise in diabetic management. In this study, we found that AMPK activation and its downstream pathways could reflect the threshold of exercise or antioxidant administration for diabetes management (*Figure 1*). This study provides clues for the personalized management of diabetes by antioxidants and exercise.

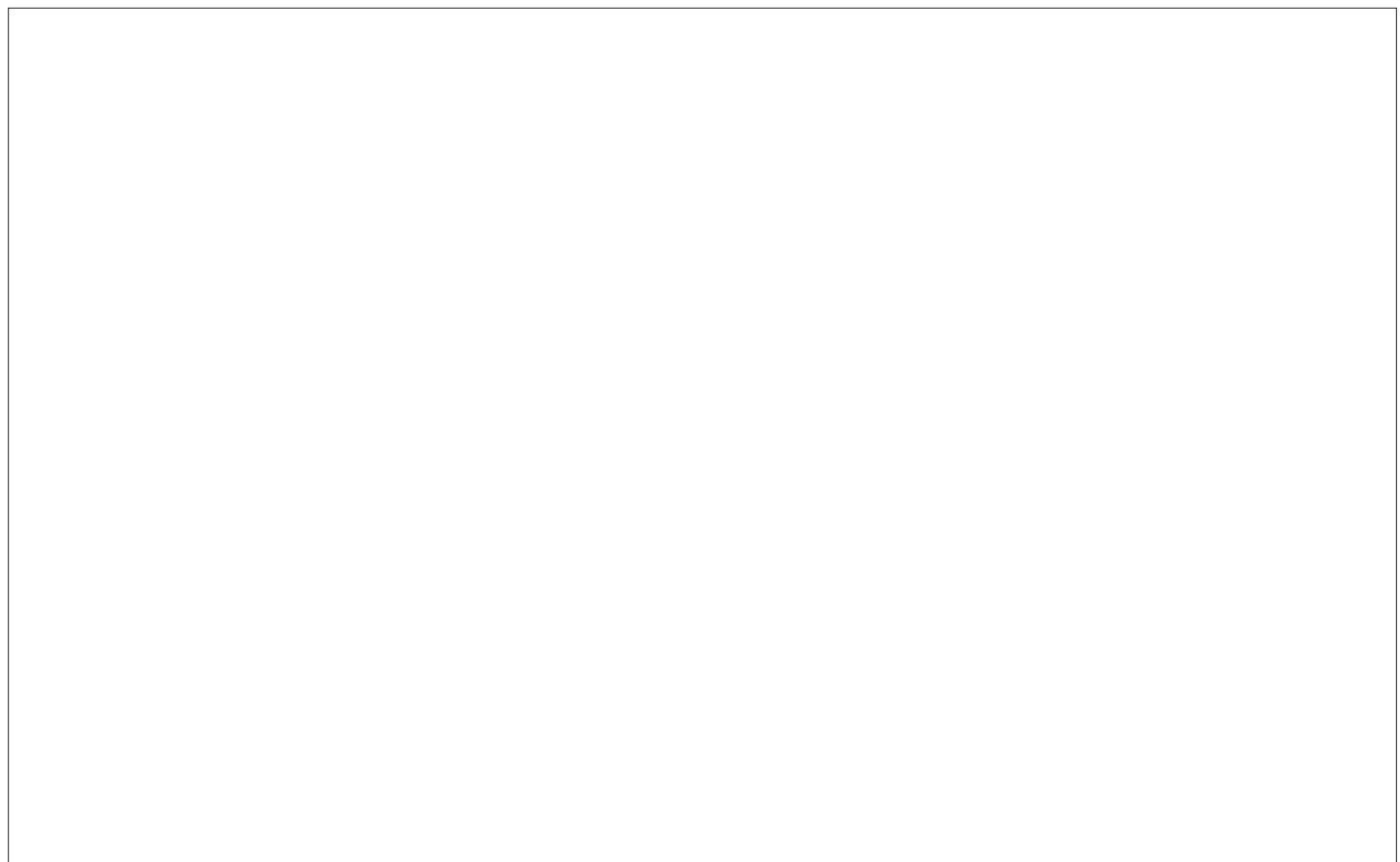

**Figure 1.** A model of the different mechanisms of exercise and antioxidant intervention in diabetes. A graphical abstract of this study. Moderate exercise upregulated compensatory antioxidant capability and reached a high-level redox balance, whereas antioxidant intervention achieved a low-level redox balance by inhibiting oxidative stress for treating diabetes. ROS: reactive oxygen species; AMPK: AMP-activated protein kinase.

## Results

### Exercise promotes antioxidant levels through producing ROS, leading to a high level of REDOX balance in the liver

To investigate the hepatic redox regulation in diabetes after exercise intervention, we established the T2DM rat model by feeding HFD followed by a low dose of STZ injection (35 mg/kg). The exercise intervention was started from day-0 to day-28 (*Figure 2A*). According to previous studies, the initial speed of exercise was 15 m/min, and the speed was increased by 3 m/min every 5 min. After the speed reached 20 m/min, the speed was maintained for another 60 min with slope of 5%. The exercise intensity was 64–76% VO$_{2max}$ (*Qin et al., 2020*). The low-intensity continuous exercise (CE) can be regarded as aerobic exercise.

The ROS-generating nicotinamide adenine dinucleotide phosphate oxidases (NOXs) have been recognized as one of the main sources of ROS production in cells (*Panday et al., 2015*). Cyclooxygenase 2 (COX2) activity could also act as a stimulus for ROS production (*Burdon et al., 2007*). The expressions of NADPH oxidase 4 (NOX4) and COX2 in the liver were increased in the diabetic group. After exercise intervention, NOX4 and COX2 level were further up-regulated compared with the diabetic group (*Figure 2B–D*). These results indicate the exercise intervention up-regulated ROS production.

Next, we detected the expression of antioxidant enzymes in liver tissue. Nuclear factor erythroid 2–related factor 2 (Nrf2) is the central regulator of the threshold mechanisms of oxidative stress and ROS generation (*McMahon et al., 2003*). With the increase of ROS level in the development of diabetes, Nrf2 was activated to induce the transcription of several antioxidant enzymes (*Bitar and Al-Mulla, 2011*; *Jiang et al., 2010*). We found an increase in Nrf2 expression in diabetic rats (*Figure 2E–F*). After

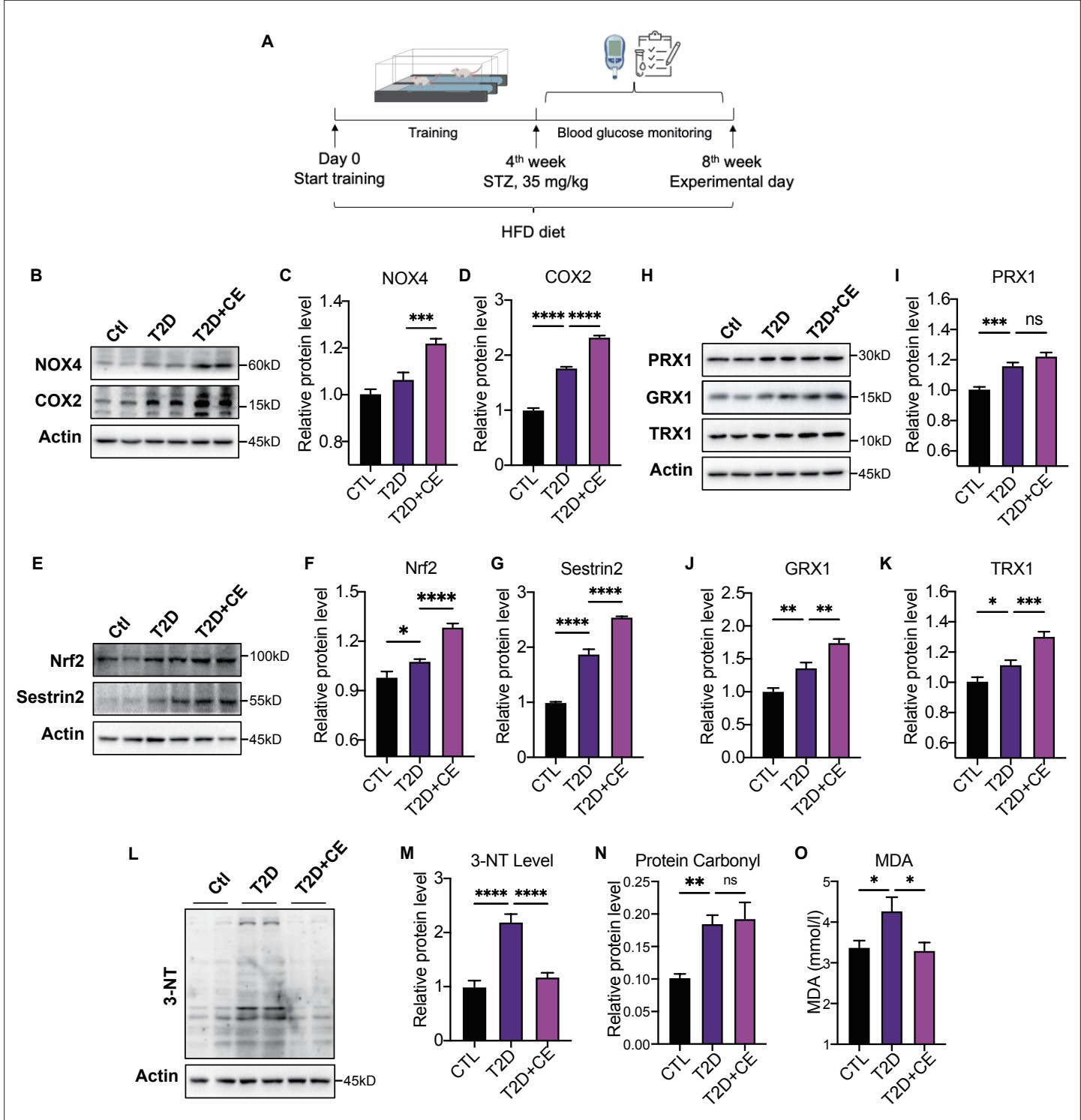

**Figure 2.** Moderate exercise induced reactive oxygen species (ROS) production in exercise group and increased the antioxidant status. (**A**) Experimental design. Type 2 diabetic model (T2DM) rats model was fed by high-fat diet plus a low dose of streptozotocin (STZ) injection (35 mg/kg). The high-fat diet (HFD, 60% calories from fat) was started from the 1st week to the 8th week. The exercise intervention was started from 1st week to 4th week. (**B–D**). Representative protein level and quantitative analysis of NADPH oxidase 4 (NOX4) (67 kDa), cyclooxygenase 2 (COX2) (17 kDa) and Actin (45 kDa) in the rats in the control (Ctl), T2D, and T2D+continuous exercise (CE) groups.(**E–G**). Representative protein level and quantitative analysis of Nrf2(97 kDa), Sestrin2 (56 kDa) and Actin (45 kDa) in the rats in the Ctl, T2D, and T2D+CE groups.(**H–K**). Representative protein level and quantitative analysis of PRX1 (27 kDa), Grx1 (17 kDa), Trx1 (12 kDa), and Actin (45 kDa) in the rats in the Ctl, T2D and T2D+CE groups. The rat livers were homogenized by 1% SDS and analyzed by Western blots with the appropriate antibodies. (**L–M**). Representative protein level and quantitative analysis of 3-NT and Actin (45 kDa)

*Figure 2 continued on next page*

*Figure 2 continued*

in the rat in the Ctl, T2D and T2D+CE groups. (**N–O**). Liver protein carbonylation (N) and MDA content (O) level was detected in the rats of Ctl, T2D, T2D+CE groups. (ns: not significant; *p<0.05, **p<0.01, ***p<0.001, ****p<0.0001 compared with all groups by one-way ANOVA and Tukey's post hoc test; data are expressed as the mean ± SEM; n=4–8 per group).

The online version of this article includes the following source data for figure 2:

**Source data 1.** Full western blot images.

**Source data 2.** Normalized grey value of western blot data.

CE intervention, the level of Nrf2 levels further increased, indicating that exercise intervention could activate antioxidant system (*Figure 2E–F*). Under stress conditions, Nrf2 translocates to the nucleus and binds to antioxidant response elements (AREs), which results in the expression of diverse anti-oxidant and metabolic genes, such as thioredoxin (Trx), to relieve oxidative damage (*Krajka-Kuźniak et al., 2017*). Thioredoxin-1 (Trx-1), a type of cytosolic isoform of Trx, has been widely accepted to regulate glutathione metabolism with GRX and PRX. After CE intervention, we found the protein expression of GRX1 and TRX1 were up-regulated (*Figure 2H J–K*). Notably, the PRX expression also showed a trend of increase (*Figure 2*). Sestrin2 is a cysteine sulfinyl reductase that plays crucial roles in regulation of antioxidant actions (*Budanov et al., 2010*). As an endogenous antioxidant, the hepatic sestrin2 level also showed a significant increase in the CE group (*Figure 2E G*).

Since excess ROS can cause the increase of oxidative damage (*Murphy et al., 2022*), we further detected the protein damage and lipid peroxidation to determine the redox status. 3-Nitrotyrosine (3-NT) and protein carbonylation are biomarkers of reactive nitrogen species (RNS) and ROS modified proteins (*Ahsan, 2013*; *Wong et al., 2010*). We found that the CE intervention reduced the 3-NT level and did not further decrease the protein carbonylation level (*Figure 2L–N*). MDA, a biomarker of lipid peroxidation, was also significantly up-regulated in the diabetic group but decreased in exercise group (*Figure 2O*). These results indicate that the high ROS production in the CE group could, instead, increase the antioxidant status to avoid oxidative damage. It suggests that CE can promote redox to reach a high level of balance. Therefore, even if exercise increases the ROS-generating enzymes NOX4 and COX2, the increase in ROS production does not lead to oxidative damage.

## Antioxidant intervention alleviates blood glucose through reducing oxidative stress, leading to a low level of REDOX balance in the liver

Recent studies have suggested that NADPH oxidase is one of the primary sources of ROS (*Panday et al., 2015*; *López-Acosta et al., 2018*). Apocynin has already been characterized as an NADPH oxidase inhibitor in the early 1980s, and it can also act as an antioxidant (*Heumüller et al., 2008*). Our previous study showed that apocynin intervention alleviated blood glucose by inhibiting oxidative products. In this study, the antioxidant supplement was also started from day-0 to day-28 in this study (*Figure 3A*). We found that apocynin supplement decreased the protein carbonylation level and MDA level in the liver (*Figure 3B–C*). Also, the total antioxidant capacity (TAOC) level was increased after apocynin treatment, indicating the decrease of oxidation level (*Figure 3D*). Moreover, as endogenous antioxidant, the Sestrin2 and Nrf2 expression decreased after apocynin intervention (*Figure 3E–G*). These results indicate that the antioxidant intervention reduced the ROS in diabetic hepatocytes, thereby decreasing the ROS-induced compensatory upregulation of Sestrin2 and Nrf2.

Consistently, Glut2, a glucose sensor in the liver, was increased in diabetic liver and decreased after the apocynin supplement (*Figure 3E, H*). The postprandial blood glucose, fasting blood glucose, and oral glucose tolerance test (2 hr after oral glucose, (oral glucose tolerance test)OGTT) were decreased in the apocynin intervention group compared with the diabetic rat group (*Figure 3K*). Consistent with the apocynin intervention group, the exercise group also showed lower postprandial blood glucose and fasting blood glucose levels and OGTT (*Figure 3K*). These studies indicate that the apocynin treatment improved the diabetes through inhibiting ROS level and protein oxidative damage to achieve a low-level redox balance.

Moderate exercise-generated ROS promotes activation of AMPK by phosphorylation and reduces blood glucose level, while excessive exercise-generated oxidative stress reduces AMPK expression and exacerbates diabetes.

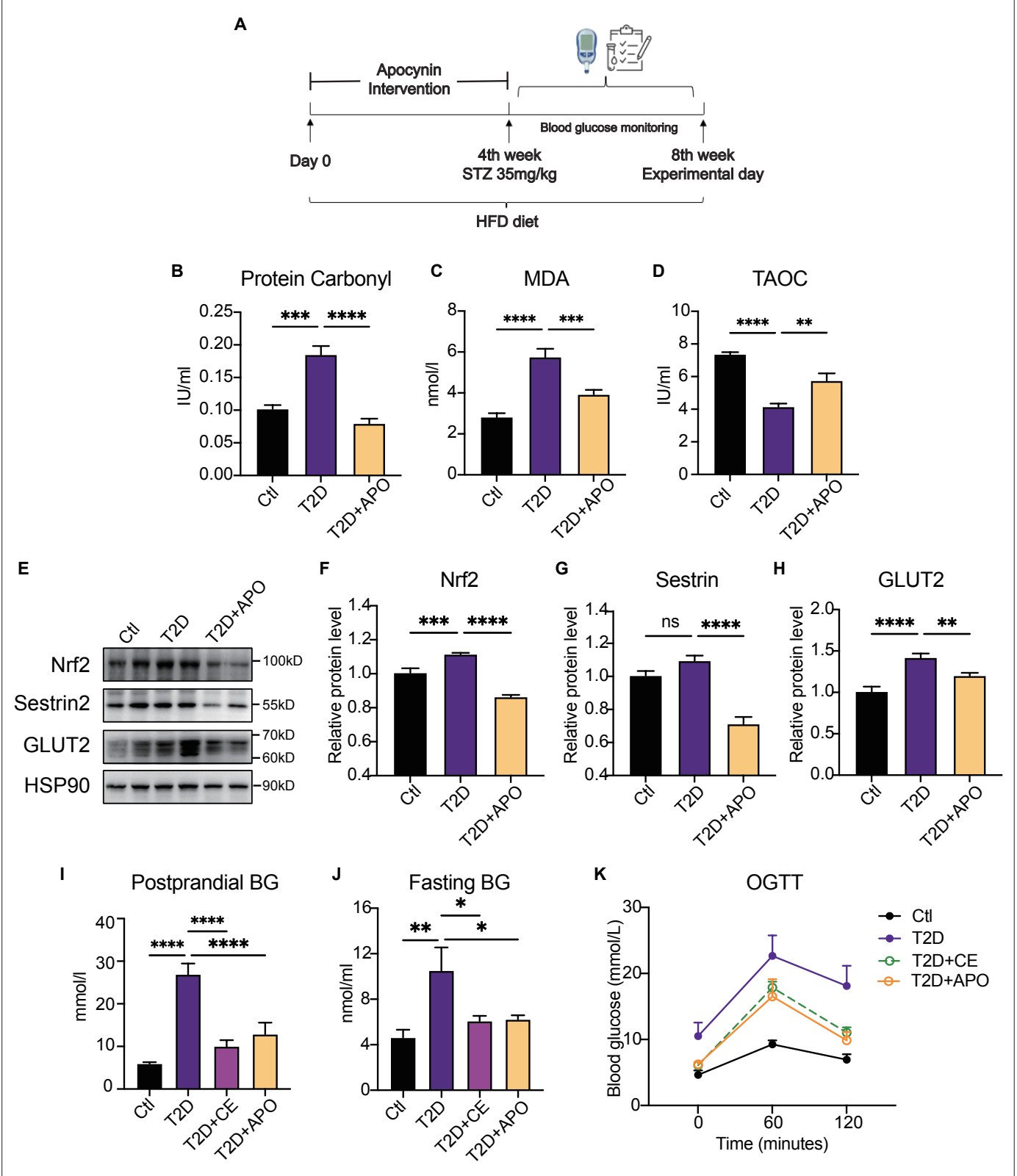

**Figure 3.** Antioxidant intervention alleviates blood glucose through promoting the upregulation of reducing levels. (**A**) Experimental design. Type 2 diabetic model (T2DM) rats model was fed by high-fat diet plus a low dose of streptozotocin (STZ) injection (35 mg/kg). The apocynin intervention was started from 1st week to 4th week. (**B–D**) Liver protein carbonylation (**B**), MDA content (**C**) and TAOC (**D**) level were detected in the rats of control (Ctl), T2D and T2D+APO groups. (**E–H**) Representative protein level and quantitative analysis of Nrf2 (97 kDa), Sestrin2 (57 kDa), Glut2 (60–70 kDa), and

*Figure 3 continued on next page*

*Figure 3 continued*

HSP90 (90 kDa) in the rats in the Ctl, T2D and T2D+APO groups. (**I**) Postprandial blood glucose levels of Ctl, T2D, T2D+continuous exercise (CE) and T2D+APO groups at the end of 8th week. (**J**) Fasting blood glucose levels of Ctl, T2D, T2D+CE and T2D+APO groups at the end of 8th week. (**K**) Blood glucose level after oral glucose administration (0 min, 60 min, and 120 min) in Ctl, T2D, T2D+CE and T2D+APO groups at the end of 8th week (*p<0.05, **p<0.01, ***p<0.001, ****p<0.0001 compared with all groups by one-way ANOVA and Tukey's post hoc test; data are expressed as the mean ± SEM; n=4–8 per group).

The online version of this article includes the following source data for figure 3:

**Source data 1.** Full western blot images.

**Source data 2.** Normalized protein carbonylation, MDA content, TAOC, grey value of western blot data and blood glucose level.

In order to find out the biomarkers that could reflect moderate exercise to improve blood glucose control, diabetic rats were divided into short-term CE, intermittent exercise (IE), and excessive exercise (EE) according to the exercise intensity and mode (*Qin et al., 2020*). We found that the random blood glucose and 2 hr OGTT in CE and IE treated diabetic rats were decreased (*Figure 4A–B*). In contrast, EE intervention did not improve blood glucose but slightly increased random and 2 hr OGTT (*Figure 4A–B*).

We detected the increase of ROS production-related enzymes, such as NOX4 and COX2 in the EE group, indicating the highest oxidation level (*Figure 4C–E*). Next, we detected the expression of antioxidant enzymes and oxidative damage in the liver tissue of exercise-treated type 2 diabetic (T2D) rats. As shown in *Figure 4F–G*, IE intervention increased the activity of MnSOD as shown by decreased level of acetylation compared with the diabetic rats. The expression of GRX and TRX were up-regulated after CE intervention (*Figure 4F H–I*). Furthermore, we detected the oxidative damage in these three modes. The results showed that the CE and IE group did not obviously change the protein carbonylation level. However, the EE intervention promoted the protein carbonylation in the liver, indicating this mode of action is not due to free radical scavenging but oxidative damage (*Figure 4J*). In addition to the protein damage, hepatic MDA concentration showed significant up-regulation in the diabetic group but decreased in CE and IE group (*Figure 4K*), while increased MDA in the EE group indicates oxidative damage. Among these three exercise modes, the IE group showed the lowest level of oxidation (the minor increase in NOX4 and a slight decrease in carbonylation). Although the levels of antioxidant enzymes such as GRX and TRX did not increase, the activity of MnSOD also increased significantly (*Figure 4F-G*). The reduction of MDA level also indicates IE group did not form oxidative damage (*Figure 4K*), indicating the IE group could also maintain a relatively high level of redox balance. Nevertheless, the decrease of antioxidant enzymes and increase of oxidative damage in the EE group indicates that the REDOX balance was disrupted.

Notably, the phosphorylation of AMPK showed different patterns in three kinds of exercise, among which both CE and IE intervention could promote the phosphorylation of AMPK compared to the diabetic rats (*Figure 4M–N*). EE intervention did not increase the content of AMPK phosphorylation, which might be caused by the reduction of AMPK level. Meanwhile, the ratio of AMP to ATP was detected, and exercise-activated AMPK did not exhibit AMP-dependent characteristics at this time (*Figure 4L*). These results suggest that moderate exercise-generated ROS may directly promote AMPK activation by phosphorylation without AMP upregulation and reduce blood and liver glucose levels. However, excessive exercise-generated oxidative stress reduces AMPK expression and exacerbates diabetes.

## Moderate exercise promoted glycolysis and mitochondrial tricarboxylic acid cycle and inhibited the gluconeogenesis in the liver of diabetic rats

Next, we further explored the mechanism by which inhibiting blood glucose during CE and IE intervention. Fructose-2,6-diphosphate (F-2,6-P2; also known as F-2,6-BP), which is a product of the bifunctional enzyme 6-phosphofructose 2-kinase/fructose 2,6-diphosphatase 2 (PFK/FBPase 2, also known as PFKFB2), is a potent regulator of glycolytic and gluconeogenic flux. The phospho-PFKFB2 to PFKFB2 ratio represents the glycolytic rate. A high ratio of phospho-PFKFB2:PFKFB2 leads to an increase in the F-2,6-P2 level and the allosteric activation of phosphor-fructose kinase 1 (PFK1), while a low ratio leads to a decrease in F-2,6-P2 and an increase in gluconeogenesis (*Okar et al., 2001*). The overexpression of bifunctional enzymes in mouse liver can reduce blood glucose levels by inhibiting

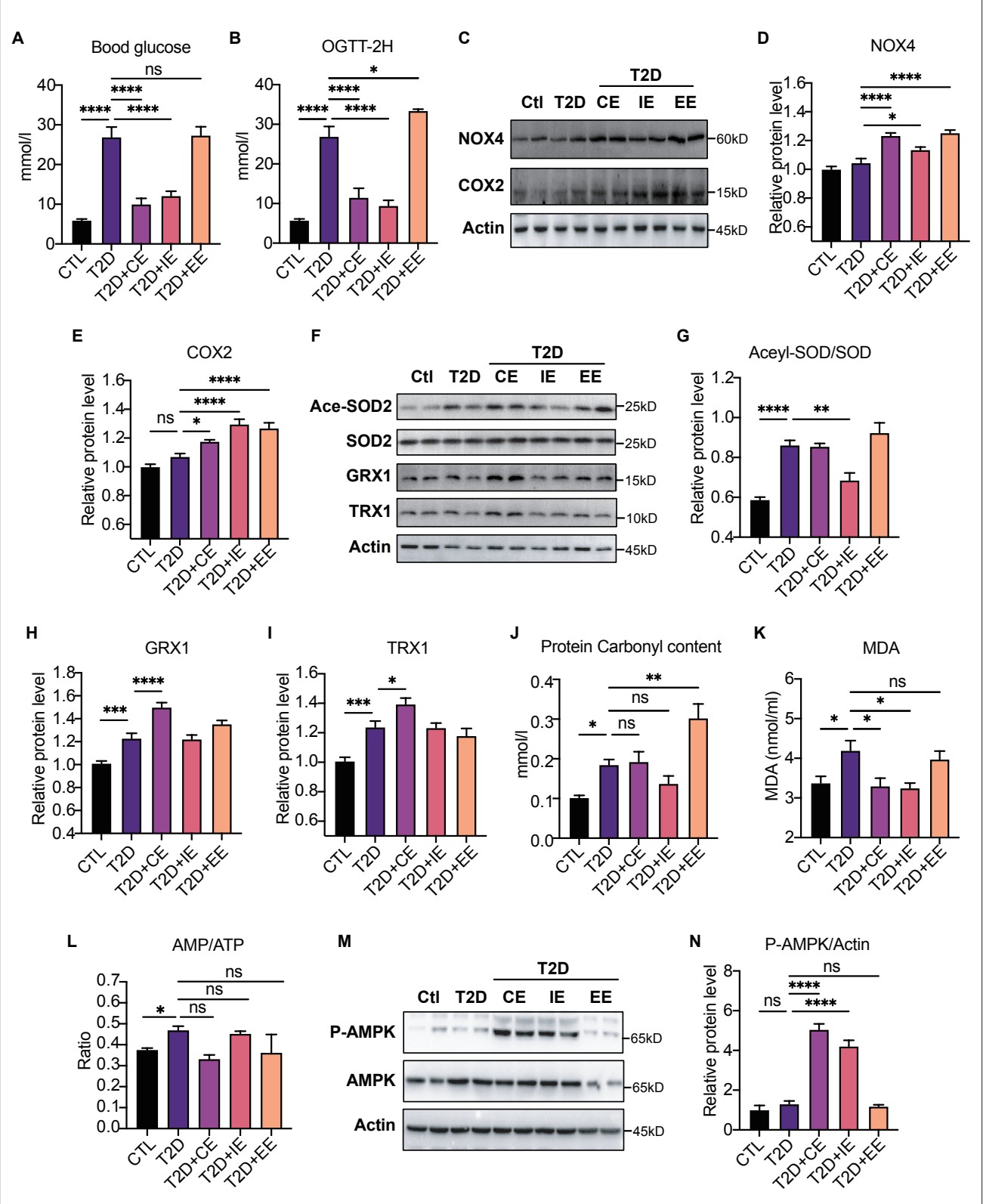

**Figure 4.** Moderate exercise-generated reactive oxygen species (ROS) promotes activation of AMP-activated protein kinase (AMPK) by phosphorylation and reduces blood glucose level, while excessive exercise- generated oxidative stress reduces AMPK expression and exacerbates diabetes. (**A**) Postprandial blood glucose levels of control (Ctl), type 2 diabetic (T2D), T2D+continuous exercise (CE), T2D+intermittent exercise (IE) and T2D+excessive exercise (EE) groups at the end of 8th week. (**B**) Blood glucose level after oral glucose administration in Ctl, T2D, T2D+CE,

*Figure 4 continued on next page*

*Figure 4 continued*

T2D+IE and T2D+EE groups at the end of 8th week. (**C–E**) Representative protein level and quantitative analysis of NADPH oxidase 4 (NOX4) (67 kDa), cyclooxygenase 2 (COX2) (17 kDa) and HSP90 (90 kDa) in the rats in the Ctl, T2D, T2D+CE, T2D+IE, and T2D+EE groups. (**F–I**) Representative protein level and quantitative analysis of Ace-SOD2 (27 kDa), SOD2 (17 kDa), Grx1 (17 kDa), Trx1 (12 kDa) and Actin (45 kDa) in the rats in the Ctl, T2D, T2D+CE, T2D+IE and T2D+EE groups. (**J–L**) Liver protein carbonylation content (J), liver MDA content (K) and AMP/ATP ratio (L) were detected in the rats of Ctl, T2D, T2D+CE, T2D+IE and T2D+EE groups. (**M**–N). Representative protein level and quantitative analysis of P-AMPK (67 kDa), AMPK (67 kDa) and Actin (45 kDa) in the rats in the Ctl, T2D, T2D+CE, T2D+IE and T2D+EE groups. (ns: not significant; *p<0.05, **p<0.01, ***p<0.001, ****p<0.0001 compared with all groups by one-way ANOVA and Tukey's post hoc test; data are expressed as the mean ± SEM; n=4–8 per group).

The online version of this article includes the following source data for figure 4:

**Source data 1.** Full western blot images.

**Source data 2.** Normalized grey value of western blot data.

hepatic glucose production (*Wu et al., 2001*). Therefore, bifunctional enzymes are also a potential target for reducing hepatic glucose production. In our study, the p-PFK2:PFK2 ratio decreased in the diabetic rats but was enhanced by CE and IE intervention (*Figure 5A–C*), suggesting that CE and IE could reverse gluconeogenesis to glycolysis by enhancing PFK/FBPase. Meanwhile, the substrates of the glycolytic pathway (such as DHAP, *Figure 5D*) and the tricarboxylic acid cycle (such as citrate, succinate, and malate, *Figure 5D*) showed an upward trend.

Besides glycolysis, gluconeogenesis is also critical in maintaining liver and blood glucose homeostasis. FoxO1 has been tightly linked with hepatic gluconeogenesis through inhibiting the transcription of gluconeogenesis-related PEPCK and G6Pase (*Haeusler et al., 2010*; *Nakae et al., 2001*). Herein, we found the expression of FoxO1 was increased in the diabetic group but reduced in the CE and IE groups (*Figure 5E–F*). Meanwhile, the mRNA level of *Pepck* and *G6PC* also decreased in the CE and IE groups (*Figure 5H–I*). These results indicate that moderate exercise (CE and IE) inhibited gluconeogenesis through down-regulating FoxO1. For the glucose uptake, we detected the protein expression of GLUT2 in the liver tissue, which helps in the uptake of glucose by the hepatocytes for glycolysis and glycogenesis. We found the level of GLUT2 was increased in diabetic rats, but down-regulated by the CE and IE intervention (*Figure 5E G*). Taken together, these results illustrated that moderate exercise promoted glucose transport, glucose catabolism and inhibited the gluconeogenesis in the liver of diabetic rats (*Figure 5J*).

## Moderate exercise inhibited hepatic mitophagy, while excessive exercise exhibited opposite effect and inhibited the mitochondrial biogenesis

The electron transport associated with the mitochondrial function is considered the major process leading to ROS production during exercise (*Zorov et al., 2014*). To further explore the downstream signal of AMPK activation in moderate and excessive exercise, we detected the protein expression of mitochondrial dynamic and mitochondrial biogenesis. According to the results in *Figure 6A–E*, we found that the mitochondrial fusion protein (MFN) was significantly decreased in the liver of the excessive exercise group, and the mitochondrial fission protein (FIS) and autophagy-related protein ATG5 and LC3B did not change, compared with the diabetic group. Notably, the ATG5 and LC3B levels decreased in the CE and IE group, compared with the diabetic group (*Figure 6A–E*). Since PGC-1α is a transcriptional coactivator and a central inducer of mitochondrial biogenesis in cells (*Austin and St-Pierre, 2012*), we measured the expression of PGC-1α and found that its expression was increased in CE and IE group, but not in the EE group (*Figure 6F*).

These results indicate that moderate exercise promoted mitochondrial biogenesis and ameliorated autophagy in the liver. However, excessive exercise aggravated mitochondrial fission and did not alleviate autophagy. Similarly, the mitochondria structure of the live tissue in the EE group was fragmented and showed greatly diminished cristae and swelling matrix under transmission electron microscopy, reflecting a defect in oxidative phosphorylation. However, the CE and IE group showed increased numbers of cristae and a clear structure of mitochondrial cristae (*Figure 6H*). These results all showed that the *in vivo* mitochondrial ROS burst caused by excessive exercise inhibited the expression of AMPK and promoted mitophagy. The damage to the mitochondrial dynamics and structure in liver tissue led to abnormal aerobic oxidation, thereby aggravating diabetes.

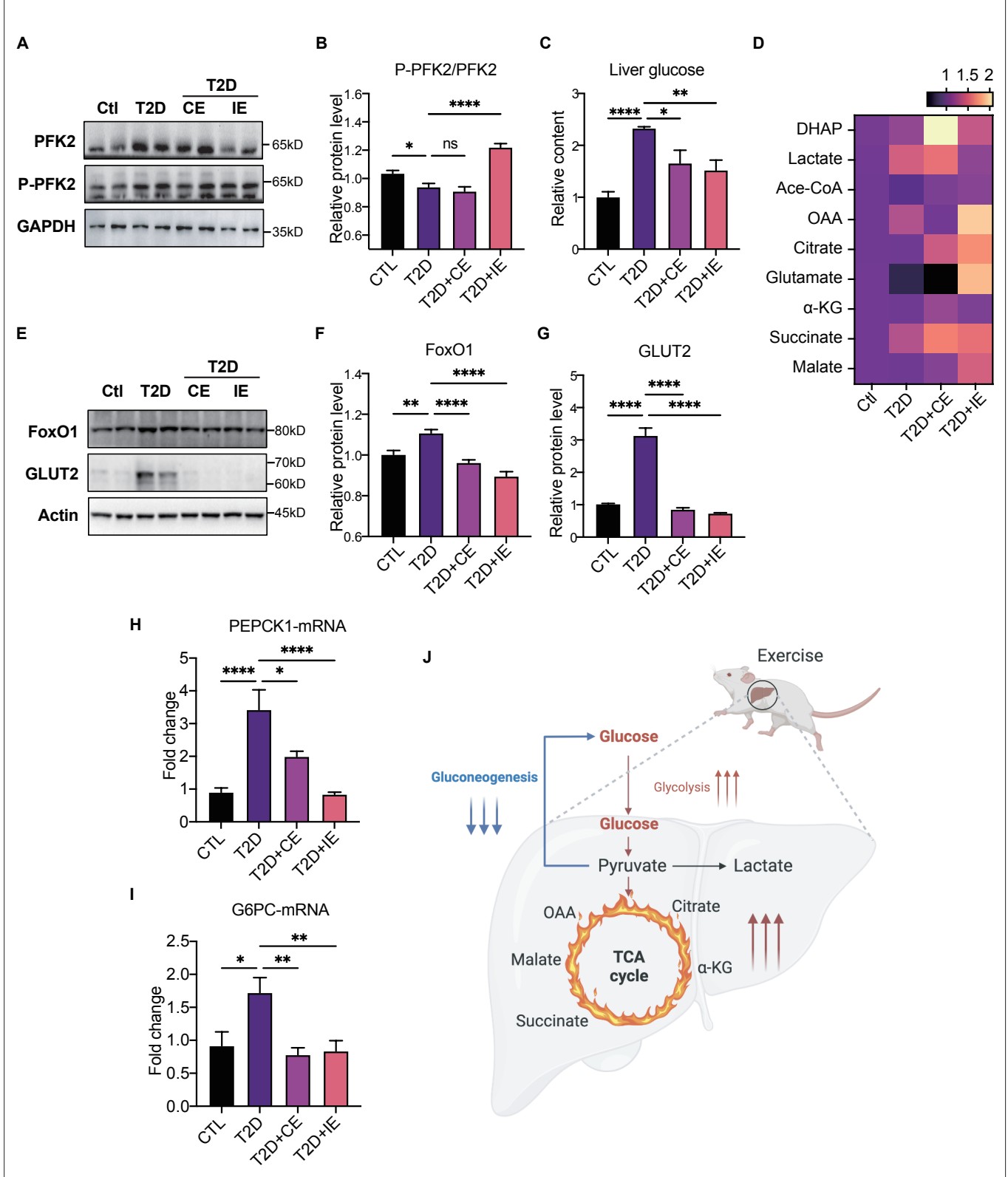

**Figure 5.** Moderate exercise promoted glycolysis and mitochondrial tricarboxylic acid cycle and inhibited the gluconeogenesis in the liver of diabetic rats. (**A–B**) Representative protein level and quantitative analysis of P-PFK2 (64 kDa), PFK2 (64 kDa) and GAPDH (37 kDa) in the rats in the control (Ctl), T2D, T2D+continuous exercise (CE) and T2D+intermitten exercise (IE) groups. (**C**). Liver glucose level after oral glucose administration in Ctl, T2D, T2D+CE, and T2D+IE groups at the end of 8th week. (**D**) Relative concentrations of substrates for glycolysis (DHAP and Lactate) and the tricarboxylic

*Figure 5 continued on next page*

*Figure 5 continued*

acid cycle (citrate, succinate and malate) in the rats of Ctl, T2D, T2D+CE, and T2D+IE groups. The concentration of substrates was analyzed by LC-MS/MS. (**E–G**) Representative protein level and quantitative analysis of FoxO1 (82 kDa), GLUT2 (60–70 kDa) and Actin (45 kDa) in the rats in the Ctl, T2D, T2D+CE, and T2D+IE groups. (**H–I**) Expression of hepatic *Pepck* and *G6C* mRNA in the Ctl, T2D, T2D+CE, and T2D+IE groups were evaluated by real-time PCR analysis. Values represent mean ratios of *Pepck* and *G6pase* transcripts normalized to GAPDH transcript levels. (**J**) Schematic diagram illustrating the effect of CE and IE on glycolysis, gluconeogenesis and mitochondrial tricarboxylic acid cycle (ns: not significant; *p<0.05, **p<0.01, ****p<0.0001 compared with all groups by one-way ANOVA and Tukey's post hoc test; data are expressed as the mean ± SEM; n=6–8 per group).

The online version of this article includes the following source data for figure 5:

**Source data 1.** Full western blot images.

**Source data 2.** Normalized grey value of western blot and mRNA data.

## Moderate ROS activates AMPK through GRX-mediated glutathionylation

To further illustrate the effect of moderate ROS on the activation of AMPK, primary hepatocytes were intervened with $H_2O_2$ to mimic ROS production *in vivo*. H₂DCFDA and dihydroethidium dyes were used to assess intracellular ROS levels in the $H_2O_2$-treated primary hepatocytes. As shown in *Figure 7A-B* , the levels of ROS (DCF), such as $H_2O_2$, and $O_2^{\bullet -}$ accumulation (dihydroethidium) were dose-dependently increased in the primary hepatocytes treated with 50–200 µmol/L $H_2O_2$. Consistent with our previous study (*Dong et al., 2016b*), we found moderate ROS would activate AMPK through GRX-mediated S-glutathionylation. As shown in *Figure 7C*, exposure to 50 and 100 µmol/L $H_2O_2$ led to an increase of GSS-protein adduct, concomitant with the AMPK phosphorylation (*Figure 7C–F*), suggesting that the ROS level within redox balance threshold could induce glutathionylation and phosphorylation of AMPK to activate AMPK. However, when the concentration of ROS was higher (200 µmol/L $H_2O_2$), both of AMPK glutathionylation and phosphorylation were decreased *Figure 7E-H*. These results indicate that activation of AMPK by moderate ROS might be mediated through GRX-mediated S-glutathionylation.

Meanwhile, we also detected the substrates of glycolysis and aerobic oxidation at different concentrations of $H_2O_2$. We found the exposure to 20–100 µmol/L $H_2O_2$, which made cells within the redox balance threshold, showed a trend of increase on glycolysis and aerobic oxidation substrates, indicating the increase of hepatic glucose catabolism (data not shown).

## Discussion

It has been shown that both antioxidants and exercise can be beneficial in substantially ameliorating hyperglycemia through ROS-mediated mechanisms in diabetes patients. However, antioxidant intervention reduces oxidative stress, while exercise produces ROS. It is imperative to explore the mechanisms underlying these seemingly paradoxical approaches for effective diabetes management.

The remission of diabetes by antioxidants intervention has been well-documented. Some compounds in food that have substantial antioxidant activities or inhibit NADPH oxidase, such as polyphenols and flavonoids (*Nie and Cooper, 2021*), have been shown to improve blood glucose and relieve type 2 diabetes in animal experiments. Several clinical trials also demonstrated the relief of diabetic hyperglycemia by antioxidants (*Movahed et al., 2020*; *Raimundo et al., 2020*). Our previous study found that hepatic mitochondrial ROS scavenger and antioxidant substances inhibited the oxidative products such as MDA and 4-HNE in diabetic mice and rats and improved blood glucose control (*Wu et al., 2021*). These results indicate that reducing the oxidative level of diabetic animals could treat diabetes. Hepatic AMPK regulates cellular and whole-body energy homeostasis, signals to stimulate glucose uptake in skeletal muscles, fatty acid oxidation in adipose and other tissues, and reduces hepatic glucose production (*Garcia and Shaw, 2017*; *Zhang et al., 2009*). Numerous pharmacological agents, including the first-line oral drug metformin, natural compounds, and hormones are known to activate AMPK (*Foretz et al., 2019*; *Pernicova and Korbonits, 2014*; *Shaw et al., 2005*; *Zhou et al., 2001*). Moreover, our previous study found that antioxidant intervention in diabetic rats could promote the phosphorylation and activation of AMPK protein, thereby regulating hepatic glucose metabolism (*Dong et al., 2016a*). Taken together, we found that the activation of AMPK

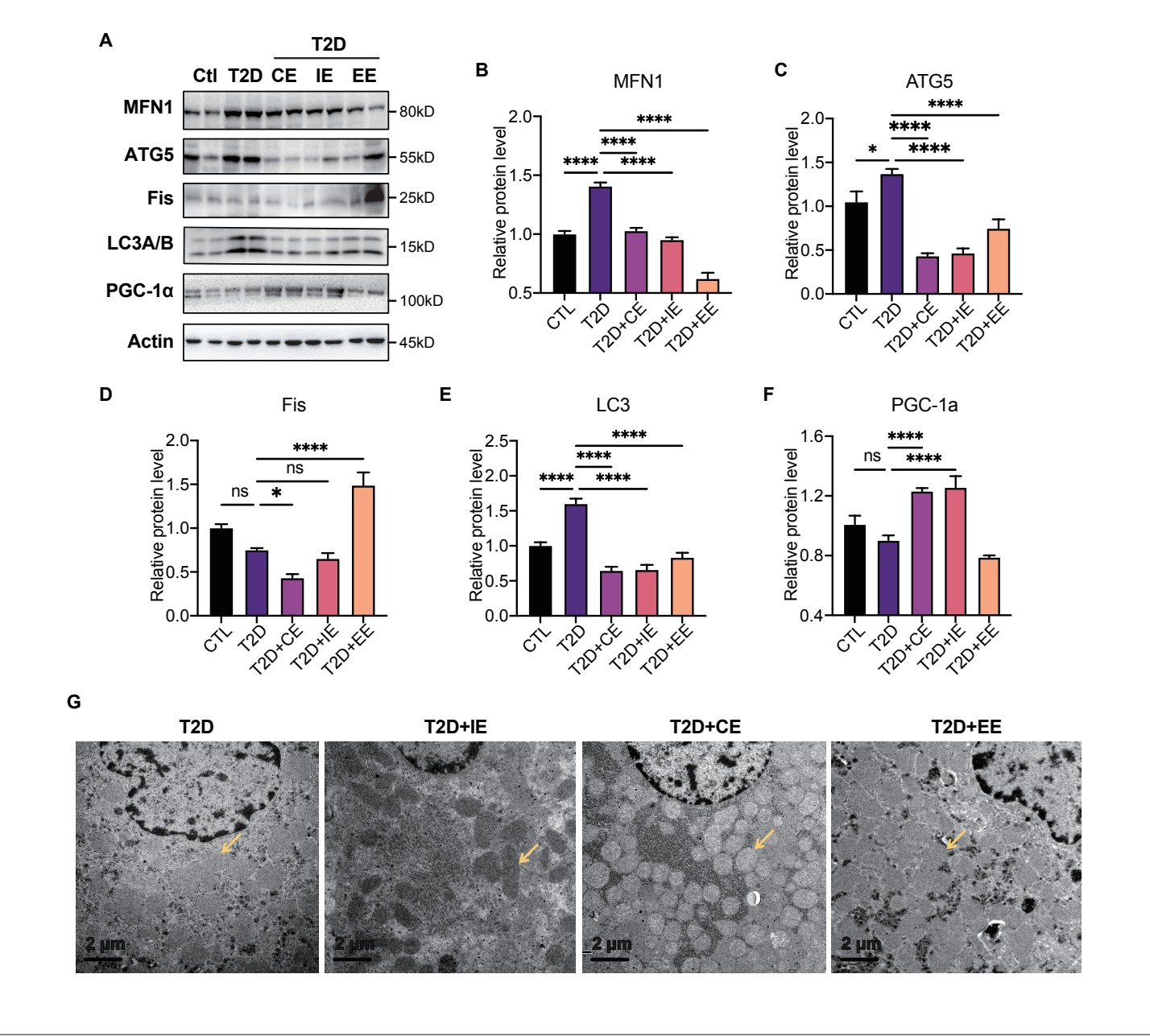

**Figure 6.** Moderate exercise inhibited hepatic mitophagy, while excessive exercise exhibited opposite effect and inhibited the mitochondrial biogenesis. (**A–F**) Representative protein level and quantitative analysis of MFN1 (82 kDa), ATG5 (55 kDa), FIS (25 kDa), LC3A/B (14,16 kDa), PGC-1α (130 kDa), and Actin (45 kDa) in the rats in the control (Ctl), T2D, T2D+continuous exercise (CE), T2D+IE, and T2D+excessive exercise (EE) groups. (**G**) Transmission electron microscope (TEM) analysis of the ultrastructure of hepatocytes in the rats in the T2D, T2D+CE, T2D+IE, and T2D+EE groups (The yellow arrows point to mitochondria). (Scale bar = 2 μm; ns: not significant; *p<0.05, **p<0.01, ***p<0.001, ****p<0.0001 compared with all groups by one-way ANOVA and Tukey's post hoc test; data are expressed as the mean ± SEM; n=8 per group).

The online version of this article includes the following source data for figure 6:

**Source data 1.** Full western blot images.

**Source data 2.** Normalized grey value of western blot data.

by antioxidant intervention was accompanied by a decrease in oxidative stress level in diabetic rats, resulting in a low level of redox balance to benefit diabetic hyperglycemia.

In the meaning time, regular exercise has been recommended to mitigate symptoms of many diseases, including psychiatric, neurological, metabolic, cardiovascular, pulmonary, musculoskeletal,

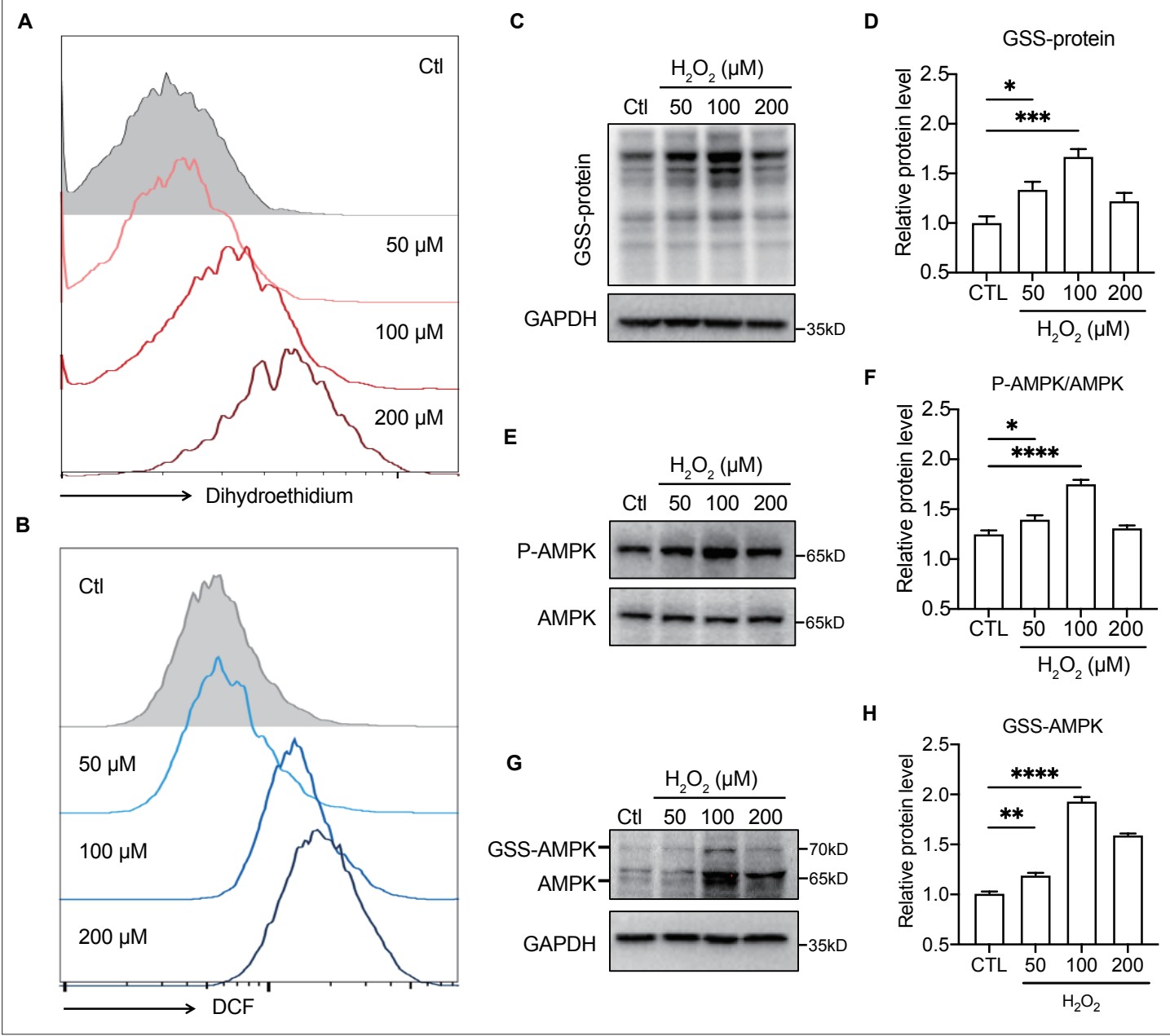

**Figure 7.** Moderate reactive oxygen species (ROS) activates AMP-activated protein kinase (AMPK) through GRX-mediated glutathionylation. (**A–B**) Analysis of superoxide (**A**) and ROS (**B**) generation using hydroethidine (**A**) and H$_2$DCFDA (**B**) probes in primary hepatocytes under H$_2$O$_2$ stress (50–200 μmol/L, 30 min). The fluorescence intensity was detected by flow cytometry. (**C–D**) Representative protein level and quantitative analysis of GSS-adduct protein and GAPDH in primary hepatocytes under H$_2$O$_2$ stress (50–200 μmol/L). Hepatocytes loaded with EE-GSH-biotin were incubated with/without H$_2$O$_2$ for 30 min, and the amounts of GSS-protein adduct formation were determined using non-reducing SDS-PAGE and Western blot analysis with streptavidin-HRP. (**E–F**) Representative protein level and quantitative analysis of P-AMPK (67 kDa) and AMPK (67 kDa) in primary hepatocytes under H$_2$O$_2$ stress (50–200 μmol/L, 30 min). (**G–H**). AMPK cysteine gel shift immunoblot. Cysteine dependent shifts by incubation of AMPK protein with glutathione reductase and PEG-Mal. PEG2-mal labelled glutathionylation modification shifts AMPK by ~10 kDa above the native molecular weight. Representative protein level and quantitative analysis of GSS-AMPK (72 kDa), AMPK (67 kDa) and GAPDH in primary hepatocytes under H$_2$O$_2$ stress.

The online version of this article includes the following source data for figure 7:

**Source data 1.** Full western blot images.

**Source data 2.** Normalized grey value of western blot data.

and even cancer (*Luan et al., 2019*). John Holloszy's studies found that exercise improved insulin sensitivity in patients with type 2 diabetes and provided a better understanding of how muscle adapts to endurance exercise (*Holloszy, 2005*; *Holloszy et al., 1986*; *Greiwe et al., 1999*; *Kirwan et al., 2009*; *Rogers et al., 1988*; *Hansen et al., 1998*). Although the benefits of exercise are irrefutable, excessive exercise is harmful (*Flockhart et al., 2021*), suggesting the importance of the amount and intensity of exercise. Recently, Chrysovalantou et al. found that NOX4 is a crucial exercise-related protein to regulate adaptive responses and prevent insulin resistance (*Xirouchaki et al., 2021*). We found that exercise could indeed increase NOX4 expression, but NOX4 was also upregulated in excessive exercise. Although Chrysovalantou's study highlights the role of the redox environment in exercise, the biomarkers of effective intervention of moderate exercise on diabetes remain unclear.

The mechanisms of exercise for the management of diabetes have previously been studied mainly around the skeletal muscle, as the activation of AMPK in skeletal muscle during exercise is considered mainly caused by the increase of intracellular AMP:ATP ratio and phosphorylation of Thr172 on the 'activation loop' 7 of the α-subunit (*Coughlan et al., 2014*). The activation of AMPK leads to the inhibition of mTORC1 activity and activation of PGC-1α, which enhances mitochondrial biogenesis and further increases muscle uptake of glucose from the blood (*Hawley et al., 2014*). However, liver energy state also plays an essential role in the activation of AMPK (*Camacho et al., 2006*) and liver is known to be critical in whole-body glucose tolerance (*Warner et al., 2020*). In addition, AMPK is also considered as a redox-sensitive protein, and its cysteine 299 and 304 sites are likely to be regulated by the oxidation of hydrogen peroxide (*Zmijewski et al., 2010*; *Hinchy et al., 2018*; *Shao et al., 2014*). Thus, AMPK might be activated not only by the increase of AMP, but also by phosphorylation through ROS regulation during exercise.

Currently there is no appropriate biomarkers to differentiate moderate and excessive exercises. According to the exercise intensity and mode, we divided the exercise groups into three modes: CE, IE for moderate exercises and EE for excessive exercise. Our study, for the first time, found that hepatic AMPK activation could act as a biomarker of dynamic redox balance during exercise to improve glycemic control in diabetic rats. ROS generated by different exercise intensities could profoundly alter the cellular redox microenvironment and directly regulate the activity and expression of hepatic AMPK through a redox-related mechanism. Moderate exercise produced optimal ROS directly promoted AMPK activation via glutathionylation in hepatocytes. The activated AMPK signaling pathway can phosphorylate and activate PFK-2 to promote glycolysis and aerobic oxidation (*Figure 8A*). Furthermore, AMPK activation can phosphorylate and inhibit the CRCT2 and class IIa HDACs pathways in the liver, thus affecting the binding of class IIa HDACs to the FOXO-family of transcription factors (*Mihaylova and Shaw, 2011*; *Mihaylova et al., 2011*). Since the gluconeogenesis related mRNA expression of PEPCK and G6Pase was mainly transcribed by FOXO1, the reduction of FOXO transcriptional activity induced by AMPK indicates the inhibition of gluconeogenesis. Therefore, under moderate exercise, the metabolic balance between glycolysis and gluconeogenesis is regulated through the activation of AMPK phosphorylation, which promotes glucose catabolism and inhibits gluconeogenesis. However, excessive ROS inhibits the activity and expression of AMPK in hepatocytes cells, which might be related with the oxidative stress induced protein degradation (*Figure 8A*). In addition, the number of mitochondria and the function of aerobic oxidation in the EE group was significantly lower than those in the moderate exercise group. In the EE group, the autophagy and fission of liver mitochondria were also up-regulated as compared with the moderate exercise, accompanied by the increase of MDA, an indicator of lipid peroxidation damage. These results indicate that AMPK signaling and oxidative damage-related index could be investigated as sensitive biomarkers for redox status changes during exercise intervention in diabetic rats. This finding can be applied to analyze the thresholds of redox balance that discriminates moderate and excessive exercise.

Exercise is considered to improve blood glucose by promoting ROS levels, which seems to be contradictory to antioxidant interventions of inhibiting ROS. Zsolt Radak et al proposed a bell-shaped dose-response curve between normal physiological function and level of ROS in healthy individuals, and suggested that moderate exercise can extend or stretch the levels of ROS while increases the physiological function (*Radak et al., 2017*). Our results validated this hypothesis and further proposed that moderate exercise could produce ROS meanwhile increase antioxidant enzyme activity to maintain high level redox balance according to the bell-shaped curve, whereas excessive exercise would

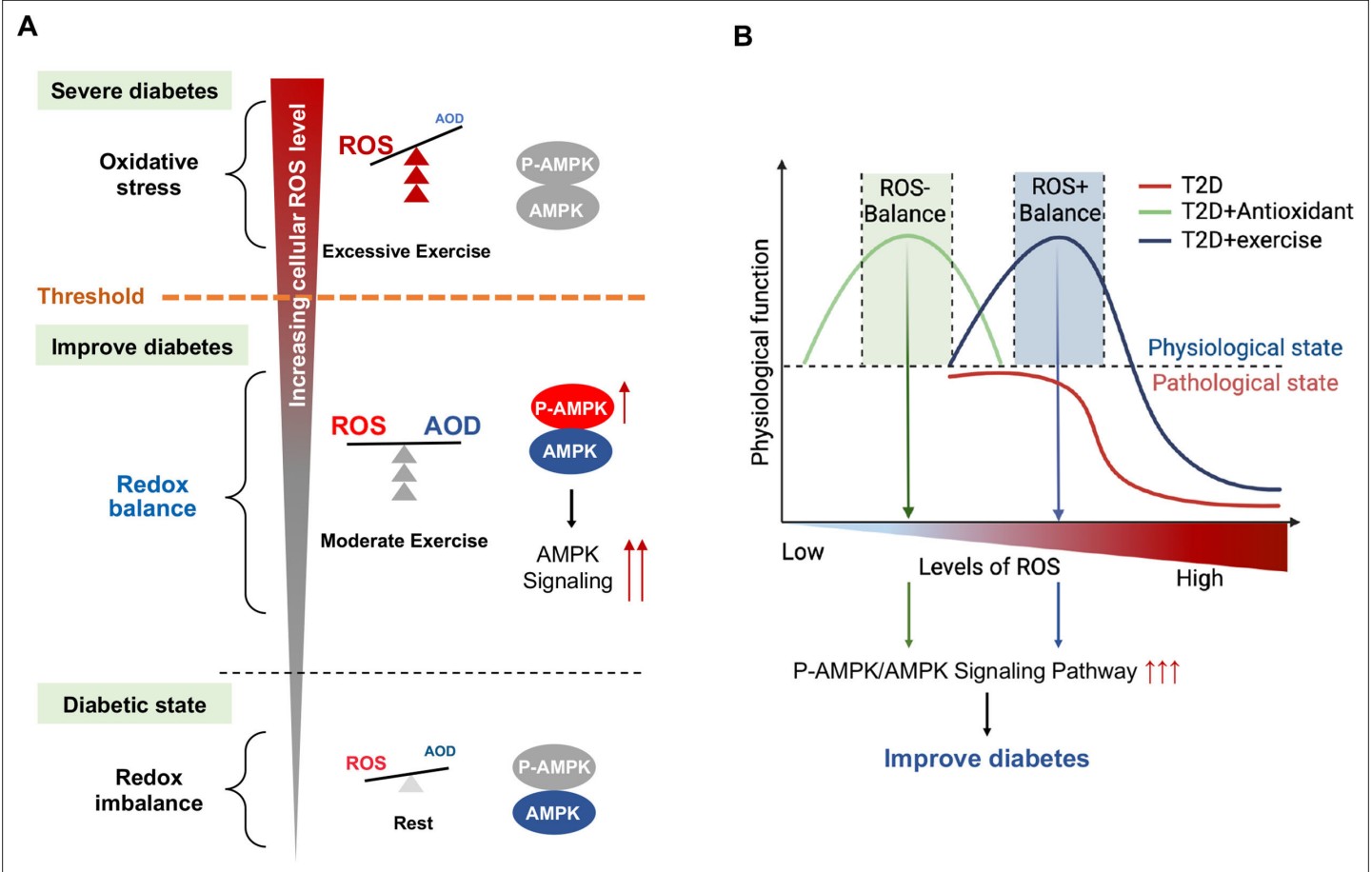

**Figure 8.** Schematic diagram of redox balance threshold. (**A**) The increased reactive oxygen species (ROS) and/or decreased antioxidative capacity (AOD) cause an imbalanced redox state and declined AMP-activated protein kinase (AMPK) activity in diabetic individuals. Moderate exercise promotes the activity of antioxidant enzymes by generating benign ROS to reach redox balance, and directly promotes AMPK signaling, thus reducing glucose levels in the blood and liver. Excessive exercise causes excess ROS and exceeds the redox balance threshold, inhibiting AMPK activity and expression, thus leading to exacerbation of diabetes. (AMPK and P-AMPK in gray circles indicate decrease, red circles indicate increase, blue circles indicate no significant changes). (**B**) Dose-response curve of diabetic individuals with exercise and antioxidant intervention.The state of diabetic individuals is applicable to the description of a S-shaped curve, due to the high level of oxidative stress and decreased reduction level in diabetic individuals. With the increase of ROS, the physiological function of diabetic individuals gradually decreases. Moderate exercise shifts the S-shaped curve upward and to the right, forming a bell-shaped dose-response curve, thus reducing the sensitivity to oxidative stress in diabetic individuals and restoring redox homeostasis. However, with excessive exercise, ROS production increases beyond the threshold range of redox balance, resulting in decreased physiological function. The ROS at the peak of the bell-shaped curve for antioxidant interventions (optimal physiological activity) is lower than the ROS at the peak for moderate exercise. The intervals on either side of the peak correspond to the range of redox balance thresholds, where antioxidant interventions are at low levels of redox balance and exercise is at high levels of redox balance.

generate a higher level of ROS, leading to reduced physiological function. In this study, we found the state of diabetic individuals is more applicable to the description of a S-shaped curve, due to the high level of oxidative stress and decreased reduction level in diabetic individuals (***Figure 8B***). With the increase of ROS, the physiological function of diabetic individuals gradually decreases. Moderate exercise shifts the S-shaped curve into a bell-shaped dose-response curve, thus reducing the sensitivity to oxidative stress in diabetic individuals and restoring redox homeostasis. However, with excessive exercise, ROS production increases beyond the threshold range of redox balance, resulting in decreased physiological function (***Figure 8B***, see the decreasing portion of the bell curve to the right of the apex).

Nevertheless, the antioxidant intervention increased physiological activity by reducing ROS levels in diabetic individuals, restoring a bell-shaped dose-response curve at low level of ROS (***Figure 8B***). Therefore, redox balance could be achieved either at low level of ROS mediated by antioxidant intervention or at high level of ROS mediated by moderate exercise, both of which were regulated by

AMPK activation. Therefore, both high and low levels of redox balance can lead to high physiological function as long as they are in the redox balance threshold range. Then, the activation of AMPK is an important sentinel biomarker of exercise or antioxidant intervention to obtain redox dynamic balance which helps restore physiological function. Therefore, personalized intervention with respect to redox balance will be crucial for the effective management of diabetes patients. For example, we speculate that patients with mild diabetes may benefit from exercise or antioxidant supplements as an appropriate treatment, while patients with severe diabetes may benefit from antioxidant supplements, rather than exercise, to help reduce their excessive levels of ROS. Meanwhile, the antioxidant intervention based on moderate exercise might offset the effect of exercise, but antioxidants could be beneficial during excessive exercise. The human study also supports that supplementation with antioxidants may preclude the health-promoting effects of exercise (*Ristow et al., 2009*). Thus, the intensity of exercise needs to be precisely adjusted to the patient's physical condition by tracking the dynamics of the redox state during exercise.

Together, our study reveals the seemingly contradictory ROS-related mechanisms of antioxidant intervention or moderate exercise for the management of diabetes. Moderate exercise promoted hepatic ROS production and up-regulated antioxidant capability, achieving a high-level balance of redox state. In contrast, antioxidant intervention scavenged the hepatic free radical to form a delicate low-level balance of redox state. Moreover, excessive exercise led to redox imbalance due to excess ROS levels. Hepatic AMPK signaling activation could act as a sign and hallmark of moderate exercise and dynamic redox balance to guide appropriate exercise or antioxidant intervention (*Figure 1*). These results illustrate that it is necessary to develop a moderate exercise program according to the REDOX microenvironment of diabetes patients. This study provides theoretical evidence for the precise management of diabetes or other metabolic diseases by antioxidants and exercise.

## Materials and methods

### Materials

TAOC kit was supplied by Changzhou Redox Biological Technology Corporation (Jiangsu, CN). Antibodies against Actin, Acetylated-Lysine, P-PFK2, Ace-SOD2, ATG5, LC3A/B, GAPDH, MFN1, and IgG-HRP were purchased from Cell Signaling Technology (USA). Antibodies against CAT, PRX1, AMPKa1, GRX1, GRX2, SOD2, HSP90, COX1, COX2, and PFK2 were purchased from ProteinTech (Wuhan, CN). Antibodies against 3-NT, 4HNE, NOX4, and PGC-1α were purchased from Abcam. Antibodies against p-AMPKα1/α2 were purchased from SAB (Signalway Antibody, USA). The detailed antibody information is shown in *Supplementary file 1a*. .

### Animal

Male SD rats (150–160 g body weight, 6–8 weeks) were purchased from Fudan University Animal Center (Shanghai, China). Normal chow and high-fat diet (HFD) were purchased from Shanghai SLRC laboratory animal Company Ltd (Shanghai, China) and the nutritional composition was shown in *Supplementary file 1b*. All animal care and experimental procedures were approved by the Fudan University Institutional Laboratory Animal Ethics Committee (NO. 20170223–123). Animals were housed in a pathogen free environment with 12 hr dark/light cycles.

### Establishment of diabetic rat model

Rats were divided into six groups in a non-blinded, randomized manner: Control (Ctl), STZ +HFD diabetic rat (T2D), Continuous exercise +STZ + HFD diabetic rat (T2D+CE), intermittent exercise +STZ + HFD diabetic rat (T2D+IE), excessive exercise +STZ +HFD diabetic rat (T2D+EE), and Apocynin +STZ + HFD diabetic rat (T2D+APO) (n=8 per group). The sample size was calculated according to the Power Curve. The diabetic rats model was established by 12 hr-fasting followed by intraperitoneal injection of 0.1 M streptozotocin (STZ) citrate solution (pH 4.5) at a dose of 35 mg/kg for day-1, and 35 mg/kg for day-2 at the 5th week. The HFD was started from the 1st week to the 8th week. After 8 weeks of intervention, the rats were sacrificed. The tissues and plasma were collected and preserved at −80°C for further analysis.

Rats were acclimated to treadmill running for 3 days before the initiation of the experiments and the exercise training intervention was continued for 4 weeks (5 times per week). All animals were randomized before the initiation of exercise tests.

## Continuous exercise

The initial speed was 15 m/min, and the speed was increased by 3 m/min every 5 min. After the speed reached 20 m/min, the speed was maintained for another 60 min with slope of 5%. The exercise intensity was 64–76% $VO_{2max}$ (*Qin et al., 2020*; *Bedford et al., 1979*; *Shepherd and Gollnick, 1976*).

## Intermittent exercise

The initial speed was 15 m/min, and the speed was increased by 3 m/min every 5 min. After the speed reached 20 m/min, the speed was maintained for 20 min and then 5 min rest at 5 m/min. The training was continued for 3 times, and the total running time is 60 min with two 5 min rest with slope of 5%.

## Excessive exercise

The initial speed was 15 m/min, and the speed was increased by 3 m/min every 5 min. After the speed reached 50 m/min, the speed was maintained for another 60 min with slope of 5%. The exercise intensity was higher than 80% $VO_{2max}$.

OGTT was performed in the fasting rats with intraperitoneal injection of glucose at 1 g/kg of body weight, and glucose was measured at 15 min, 30 min, 60 min, and 120 min, respectively. Blood glucose was determined by glucometer (Roche, Switzerland).

## Cell culture

Normal Human Hepatic Cell Line L02 cells (Cell Bank of Chinese Academy of Sciences) were grown in DMEM supplemented with 10% FBS (GIBCO, USA) in a humidified incubator (Forma Scientific) at 37°C and 5% $CO_2$ as described previously. The medium were supplemented with 10% FBS (GIBCO, USA), 2 mmol/l glutamine, 1 mmol/l sodium pyruvate, 10 mmol/l HEPES, 50 µmol/L β-mercaptoethanol, 105 U/l penicillin and streptomycin. Glutamine and sodium pyruvate were purchased from Sinopharm Chemical Reagent Co., Ltd, HEPES were purchased from Beyotime Biotechnology (Shanghai, CN). All cell lines used in the study were tested for mycoplasma and were STR profiled.

## Flow cytometry

For measurement of intracellular Superoxide, primary hepatocytes were stained with 5 µmol/L hydroethidine (superoxide indicator) (Thermo Fisher Scientific, USA). Stained cells were analyzed with NovoCyte Quanteon flow cytometer (Agilent Technologies, Inc), and acquired data were analyzed with NovoExpress software (Agilent Technologies, Inc) and FlowJo software (TreeStar, Ashland, OR).

## ATP and AMP content analysis

Liver tissue (20–30 mg) were homogenized on ice by perchloric acid. Homogenized samples were centrifuged for 12,000 rpm at 4°C (30 min). Supernatant was then neutralized with 4 M $K_2CO_3$, followed by further centrifugation for 12,000 rpm at 4°C for 20 min. Supernatant was obtained for the determination of ATP and AMP content by high performance liquid chromatography (HPLC). The detection wavelength was 254 nm.

## SOD activity and MDA content assay

SOD activity assay was carried out using a chemiluminometric detector (Lumat LB9507, Berthold). Superoxide anions were generated by adding xanthine oxidase (XO) into the reaction system consisting of xanthine and Lucigenin. The drop of luminescence within 2 min was recorded as the relative SOD activity. MDA from the oxidative polyunsaturated fatty acids (PUFA) degradation was determined as described previously (*Wu et al., 2019*).

## S-Glutathionylation of AMPK and detection of GSS-AMPK adduct formation

GSS-AMPK adduct were measured as described previously (*Zmijewski et al., 2010*; *Dong et al., 2016b*). Primary hepatocytes ($2 \times 10^6$ cell/well) were incubated with ethyl ester GSH-biotin (6 mM) for

1 hr. Cells were then washed twice with culture buffer to remove the excess of GSH and treated with $H_2O_2$ for 30 min. Cell lysates were prepared in the presence of N-ethylmaleimide (5 mM) and then passed through Bio-Gel P10 to remove free GSH-biotin and N-ethylmaleimide. The level of GSS-protein conjugates was determined using non-reducing Western blot analysis with streptavidin-HRP, whereas GSS-AMPK subunit levels were measured after pull-down with streptavidin-agarose (60 min at 4°C), followed by reducing SDS-PAGE and Western blot analysis with antibodies against AMPK α subunit.

## Metabolite profiling detection

Cellular metabolites were extracted and analysed by LC-MS/MS. Ferulic acid was added as an internal standard to metabolite extracts, and metabolite abundance was expressed relative to the internal standard and normalized to cell number. Mass isotopomer distribution was determined by LC-MS/MS (AB SCIEX Triple-TOF 4600) with selective reaction monitoring (SRM) in positive/negative mode.

## Transmission electron microscope (TEM)

Rat liver tissue (1 mm×1 mm) was fixed by paraformaldehyde. The samples were examined with a Jeol Jem-100SV electron microscope (Japan) which was operated at 80 Kv after fixed by 3% glutaraldehyde in 0.1 M phosphate buffer (pH 7.3) at Institute of Electron microscopy, Shanghai Medical College of Fudan University.

## Western blot analysis

Cells were lysed RIPA buffer with phosphatase inhibitor at 4°C for 30 min. Cell lysates were resolved by 10% SDS-PAGE, transferred to polyvinylidene fluoride (PVDF) membranes, and probed with primary antibodies. Membranes were incubated with peroxidase-conjugated secondary antibodies and visualized using a chemiluminescent substrate (ECL; GE Amersham Pharmacia, Beijing, China) and Tanon-5200 Chemiluminesent Imaging System.

## Statistics

The experimental data were expressed as mean ± SEM. One-way ANOVA was used to compare among groups. Data analysis was conducted by Graphpad prism 9 statistical analysis software. $p < 0.05$ was considered statistically significant. Data are expressed as means ± SEM; n=3 for cells experiment (n=3 represents three times of individual experiments); n=8 for animal experiments.

## Acknowledgements

The authors thank Dr. Xiaodong Zhang from Chengdu Brilliant Pharmaceuticals for his proof reading and editing of the manuscript. The authors are also indebted to Dr. Rutan Zhang and Prof. Liang Qiao from Fudan University, for analysis of LC-MS/MS data. The authors thank Dr. Ziqing Tang, Mr. Qi Kang, Dr. Xiaomin Liu, Mr. Yipei He, Dr. Xiao Zhang, Mrs Lihan Jiang, Mr. Kelei Dong for their assistance in animal experiments. The authors are also indebted to Institute of Electronmicroscopy from Fudan University, for the help on electronmicroscopy analysis. This work was supported by grants from the National Natural Science Foundation of China (Grants No. 31770916).

## Additional information

### Funding

| Funder | Grant reference number | Author |
| --- | --- | --- |
| National Natural Science Foundation of China | 31770916 | Dongyun Shi |

The funders had no role in study design, data collection and interpretation, or the decision to submit the work for publication.

## Author contributions
Meiling Wu, Conceptualization, Data curation, Formal analysis, Validation, Investigation, Visualization, Methodology, Writing – original draft; Anda Zhao, Conceptualization, Data curation, Formal analysis, Investigation, Methodology; Xingchen Yan, Xiaomin Liu, Data curation, Investigation, Methodology; Hongyang Gao, Formal analysis, Visualization, Methodology; Chunwang Zhang, Conceptualization, Data curation, Formal analysis; Qiwen Luo, Data curation, Methodology, Project administration; Feizhou Xie, Supervision, Investigation, Methodology; Shanlin Liu, Formal analysis, Supervision, Investigation, Methodology; Dongyun Shi, Data curation, Formal analysis, Supervision, Funding acquisition, Investigation, Methodology, Project administration, Writing – review and editing

## Author ORCIDs
Meiling Wu ⓘ http://orcid.org/0000-0002-2654-4896
Dongyun Shi ⓘ http://orcid.org/0000-0003-4179-1913

## Ethics
All animal care and experimental procedures were approved by the Fudan University Institutional Laboratory Animal Ethics Committee (NO. 20170223-123).

## Decision letter and Author response
Decision letter https://doi.org/10.7554/eLife.79939.sa1
Author response https://doi.org/10.7554/eLife.79939.sa2

# Additional files

## Supplementary files
• MDAR checklist
• Supplementary file 1. Antibody information (a) and chow and HFD diet nutrition composition (b).

## Data availability
Figure 1—source data 1 and 2, Figure 2—source data 1, Figure 3—source data 1 and 2, Figure 4—source data 1 and 2, Figure 5—source data 1 and 2, and Figure 6—source data 1 and 2 contain the numerical data used to generate the figures.

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
