## [Editor Report]

Redox signaling is a dynamic and concerted orchestra of interconnected cellular pathways. There is always a debate whether ROS (reactive oxygen species) could be a friend or foe. There are several paradoxical studies (both animal and human) wherein exercise health benefits were reported to be accompanied by increases in ROS generation. Utilizing the in-vitro studies as well as rats models, this manuscript illustrates the different regulatory mechanisms of exercise and antioxidant intervention on redox balance/redox state(s) that are linked to improved glucose control and thereby effective management of diabetes.

---

## [Decision Letter]

**Decision letter after peer review:**

Thank you for submitting your article "Hepatic AMPK activation in response to dynamic REDOX balance is a biomarker of exercise to improve blood glucose control" for consideration by *eLife*. Your article has been reviewed by 2 peer reviewers, and the evaluation has been overseen by a Reviewing Editor and Mone Zaidi as the Senior Editor. The following individual involved in review of your submission has agreed to reveal their identity: Muthuswamy Balasubramanyam (Reviewer #1).

Essential revisions:

*Reviewer #2 (Recommendations for the authors):*

Some concerns that require attention by the authors are as follows:

1. In Figure 1 and Figure 3, the authors detect PRX1/TRX1/GRX1, NOX/COX2, protein carbonyl, Ace-SOD/SOD and lipid peroxidation (MDA) to overview the redox state in liver tissue. It would be better to have the result of protein oxidation / nitration in liver tissue.

2. In Figure 2H-I and Figure 3A-B, it is better to also include the graph of OGTT tests with the blood glucose level plot against time instead of showing the area under curve data. In addition, the authors also recommended to implement the fasting blood glucose in relation to the disease of interest.

3. In Figure 4I, the schematic diagram did not clearly show the effect of CE and IE on promoting TCA and glycolysis. Please add the information in the revised manuscript.

4. in Figure 5G, the missing scale bar in IE, CE & EE group. And the structure mentioned in the main text should be labelled by arrows.

5. The internal reference in Figure 6 is not the same, could the authors explain the reasons or replace them with the same house-keeping protein?

6. In the methods and materials part, please clarify the detailed information on the protocol related to apocynin intervention, the methodology of detecting SOD activity, MDA content and the glutathionylation on AMPK

7. The antibodies need either clone or production numbers.

8. In the discussion part, the specific mechanism of redox balance induced AMPK activation has not been discussed in depth. Did the authors also detect the other types of oxidative modification on AMPK? The authors should discuss the potential mechanism of ROS induced AMPK activation during exercise clearly.

9. Although the paper is well-written, there are still some grammatical errors that exist throughout the manuscript. Please edit the entire text carefully in the revised manuscript.

---

## [Author Response]

Essential revisions:Reviewer #2 (Recommendations for the authors):Some concerns that require attention by the authors are as follows:1. In Figure 1 and Figure 3, the authors detect PRX1/TRX1/GRX1, NOX/COX2, protein carbonyl, Ace-SOD/SOD and lipid peroxidation (MDA) to overview the redox state in liver tissue. It would be better to have the result of protein oxidation / nitration in liver tissue.

Thanks for the suggestion. Herein, we added the result of protein nitration (3-NT) level in Figure 2L-2M in this version.

2. In Figure 2H-I and Figure 3A-B, it is better to also include the graph of OGTT tests with the blood glucose level plot against time instead of showing the area under curve data. In addition, the authors also recommended to implement the fasting blood glucose in relation to the disease of interest.

We appreciated the reviewer’s suggestion. In the revised figures, we added the fasting blood glucose and presented the line graph of OGTT in Figure 3I-3K.

3. In Figure 4I, the schematic diagram did not clearly show the effect of CE and IE on promoting TCA and glycolysis. Please add the information in the revised manuscript.

Thanks for the suggestion. According, the schematic diagram has been redesigned in Figure 5J in this version.

4. In Figure 5G, the missing scale bar in IE, CE & EE group. And the structure mentioned in the main text should be labelled by arrows.

Thanks for the suggestion. We have properly labelled the images in Figure 6G.

5. The internal reference in Figure 6 is not the same, could the authors explain the reasons or replace them with the same house-keeping protein?

Thanks for the suggestion. We understand the reviewer's concern on the internal reference of western blot. Herein, we reconducted the cell experiment in Figure 7 to convincing these results.

6. In the methods and materials part, please clarify the detailed information on the protocol related to apocynin intervention, the methodology of detecting SOD activity, MDA content and the glutathionylation on AMPK

The methods of SOD activity, MDA content and the glutathionylation of AMPK has added into the methods and materials part (Page 18-19, line 508-521).

7. The antibodies need either clone or production numbers.

The production numbers (Cat NO.) of antibodies are added in Supplementary table 1.

8. In the discussion part, the specific mechanism of redox balance induced AMPK activation has not been discussed in depth. Did the authors also detect the other types of oxidative modification on AMPK? The authors should discuss the potential mechanism of ROS induced AMPK activation during exercise clearly.

Thank you for the suggestion, we have discussed the potential mechanism of ROS induced AMPK activation during exercise in the discussion part (Page 12-13, line 363-378)

9. Although the paper is well-written, there are still some grammatical errors that exist throughout the manuscript. Please edit the entire text carefully in the revised manuscript.

We thank the reviewer’s comments. We have carefully gone through the manuscript to correct grammatical errors.